# Symbiotic and Asymbiotic Germination of *Dendrobium officinale* (Orchidaceae) Respond Differently to Exogenous Gibberellins

**DOI:** 10.3390/ijms21176104

**Published:** 2020-08-25

**Authors:** Juan Chen, Bo Yan, Yanjing Tang, Yongmei Xing, Yang Li, Dongyu Zhou, Shunxing Guo

**Affiliations:** Institute of Medicinal Plant Development, Chinese Academy of Medical Sciences & Peking Union Medical College, Beijing 100193, China; yanbo1823@gmail.com (B.Y.); yanjingtang1@gmail.com (Y.T.); ymxing@implad.ac.cn (Y.X.); 15114605567ly@gmail.com (Y.L.); zhoudy651@gmail.com (D.Z.)

**Keywords:** orchid mycorrhiza, plant hormone, symbiosis germination, gene expression

## Abstract

Seeds of almost all orchids depend on mycorrhizal fungi to induce their germination in the wild. The regulation of this symbiotic germination of orchid seeds involves complex crosstalk interactions between mycorrhizal establishment and the germination process. The aim of this study was to investigate the effect of gibberellins (GAs) on the symbiotic germination of *Dendrobium officinale* seeds and its functioning in the mutualistic interaction between orchid species and their mycobionts. To do this, we used liquid chromatograph-mass spectrometer to quantify endogenous hormones across different development stages between symbiotic and asymbiotic germination of *D. officinale*, as well as real-time quantitative PCR to investigate gene expression levels during seed germination under the different treatment concentrations of exogenous gibberellic acids (GA_3_). Our results showed that the level of endogenous GA_3_ was not significantly different between the asymbiotic and symbiotic germination groups, but the ratio of GA_3_ and abscisic acids (ABA) was significantly higher during symbiotic germination than asymbiotic germination. Exogenous GA_3_ treatment showed that a high concentration of GA_3_ could inhibit fungal colonization in the embryo cell and decrease the seed germination rate, but did not significantly affect asymbiotic germination or the growth of the free-living fungal mycelium. The expression of genes involved in the common symbiotic pathway (e.g., calcium-binding protein and calcium-dependent protein kinase) responded to the changed concentrations of exogenous GA_3_. Taken together, our results demonstrate that GA_3_ is probably a key signal molecule for crosstalk between the seed germination pathway and mycorrhiza symbiosis during the orchid seed symbiotic germination.

## 1. Introduction

Orchidaceae is among the largest families of flowering plants, one that is fascinating and rich in species diversity, with diverse pollination mechanisms and a unique mycorrhizal symbiotic relationship [1]. Orchid mycorrhizae differ from other major types of mycorrhizae in that, besides mineral nutrients (e.g., P and N), this type of fungus supplies carbohydrates to the plant, especially in the early stages of seed germination and seedling development [2]. Some orchid species are obligate symbiotic partners with fungi during their whole life cycle (e.g., mycotrophic orchid *Gastrodia* spp.), while some epiphytic and terrestrial orchids can alternately depend on fungi in their adult stages [3].

Orchid seeds are numerous, ranging from 1300 to 4,000,000 seeds per capsule, but they are extremely small and dust-like, with an undifferentiated embryo, limited storage reserves and lacking an endosperm [4]. Accordingly, seed germination and the subsequent development of the protocorm of almost all orchids is dependent either on mutualistic symbiosis with a compatible fungus, such as the member of Tulasnelloid, Sebacinaoid, and Ceratobasidiaceae under natural field conditions (symbiotic germination, SG) [5] or the replacement of the fungus by an exogenous nutrient substance in medium under controlled conditions (asymbiotic germination, AG) [6]. Thus, symbiotic germination is acknowledged as being a unique and important topic of orchid seed biology.

Morphological and cytological studies have shown that fungi enter the embryo of orchid seeds through the suspensor end, then form hyphae coil (pelotons) in the cortical cells of the embryo and finally the pelotons are digested by the embryo cell while the embryo undergoes dramatic development: from being swollen, turning light green (stage 1) to a ruptured seed coat (stage 2), then forming green protocorms (stage 3) and, finally, having expanded leaves on a developed young seedling (stage 4 and stage 5) [7,8]. Arrays of microtubules and actin microfilaments are reportedly involved in the infection droplet release and symbiosome development during legume–rhizobia interactions and establishment of arbuscular mycorrhiza, even probably in the peloton’s lysis of orchid mycorrhizae [9]. In addition, the reserve substance of the embryo also undergoes extreme changes; for example, the lipid body and protein in the embryo cell is gradually degraded, and starch grains appear at the beginning of fungal inoculation but these are gradually depleted with the symbiotic germination progress of *Dendrobium officinale* seeds [8]. All this recent research has sketched a relatively clear outline from a cell ultrastructure perspective of the symbiotic germination process of orchid seeds. Yet the molecular mechanism establishing the mycorrhizal relationship between fungi and orchid seeds at their early germination stage remains unclear.

As is well-known, plant hormones, especially gibberellins (GAs) and abscisic acids (ABA), play crucial roles not only in seed germination but also in mycorrhizal establishment. The key steps in the signal transduction pathway for GAs’ biosynthesis, metabolism, and seed germination regulation have been demonstrated clearly, and the involved genes encoding key regulatory enzymes related to GAs’ biosynthesis have been identified, such as gibberellin 20 oxidase (GA20ox) and GA3-oxidase (*GA3ox*) related to GA biosynthesis, and gibberellin 2-oxidase (GA2ox) that catalyzes the degradation of GAs [10]. DELLA proteins are reportedly central players in hormone-mediated crosstalk and they can interact through the N-terminal domain with the GA receptor encoded by GID1, through which GAs promote DELLA degradation. In addition, DELLA proteins are recognized as common components of the mycorrhizal signaling pathway and mutations to them can cause rice to fail to form mycorrhizal relationships [11,12]. Thus, we hypothesized that biosynthesis and signal transduction pathway of GAs contribute to crosstalk with the common symbiotic pathway (CSP)—a putative signal transduction pathway shared by arbuscular mycorrhizas and the rhizobium-legume symbiosis—transducing glomeromycotan or rhizobial signal perception from the plasma membrane into the nucleus during the symbiotic germination of *D. officinale* seeds [13].

To address our hypothesis, we took the *D. officinale* (an epiphytic orchid) inoculated with *Tulasnella* sp. as a model system, because *D. officinale* is among the Chinese traditional medicinal plants whose genomes have been sequenced [14], and the genome of the fungus *Tulasnella calospora* (a orchid mycorrhizal fungus) is sequenced [15]. The aims of this study were twofold. (1) To identify the differentially expressed genes (DEGs) involved in the biosynthesis and signal transduction of plant hormones and profile their expression patterns based on transcriptomic data. (2) To quantify and analyze the endogenous hormones’ level in the orchid at different germination stages between AG and SG and analyze the effect of exogenous GA_3_ upon AG and SG of *D. officinale* seeds. This study provides a new insight for better understanding orchids’ seed biology and their symbiotic mechanism and provides important data for cultivation of *D. officinale* and other medicinal orchid plants via mycorrhizal techniques.

## 2. Results

### 2.1. Determination of Endogenous Hormones’ Level at Different Germination Stages between SG and AG of D. officinale

In the previous study, we experimentally demonstrated that the seed germination of *D. officinale* on the oatmeal agar (OMA) medium with fungi is faster than seed germination on 1/2 MS medium without fungi [7]. *D. officinale* seeds usually take 10 days to develop up to stage 2 in SG, compared to 16 days in AG (Figure 1). After 2 weeks of sowing seeds, more than 50% of the seeds formed the protocorm structures (stage 3) in SG while the protocorm formation took at least 3 weeks in AG. After about 20 days, seeds in SG can develop seedling stage (stages 4), compared to 30 days in AG. It took approximately 5 weeks to finish the germination process in SG and at least two months in AG.

To understand the dynamic changes of endogenous hormones’ content during seed germination of *D. officinale*, using liquid chromatograph-mass spectrometer (LC-MS/MS), we quantified the five kinds of endogenous hormones—GA_3_, ABA, indole-3-acetic acid (IAA), *trans*-zeatin (ZT), and jasmonic acid (JA)—on the free-living fungus and the differently developed seeds in SG and AG, respectively (stage 0, no germination; stage 2, early germination; stage 3, protocorm; stage 4, seedlings) (Figure 1). These results showed that ungerminated seeds have the highest ABA content (12.78 ng/g·FW), but ABA content decreased as seed germination progressed (Table 1). The GA_3_ content rose at the early germination stage (stage 2) but declined as seeds developed. The ABA and GA_3_ contents were similar between AG and SG at the same stage, but the ratio of GA_3_/ABA was significantly higher in SG than AG (*p* < 0.05) (Figure 2). Interestingly, IAA was dramatically increased in the protocorm stage of SG (25.91 ng/g·FW) when compared to AG (0.48 ng/g·FW). This result likely explains the faster differentiation rate in the protocorm stage in SG (2 weeks) than AG (3 weeks) during *D. officinale* seed germination. Additionally, minute amounts of ZT (0.0075~0.014 ng/g·FW) were detected in both the free-living mycelium of fungus and ungerminated seeds. For ungerminated seeds, JA could not be detected and the free-living mycelium of *Tulasnella* sp. (S6) featured a low JA content (1.63 ng/g·FW), but JA peaked most in the early germination stage (stage 2) in AG (Table 1). Further, all five kinds of hormones were detected in free-living mycelium of mycorrhizal fungus *Tulasnella* sp., albeit their context ranged almost 10-fold (0.44~4.29 ng/g·FW) (Table 1).

### 2.2. Effect of Exogenous GA_3_ Treatment on Phenotypic Changes in D. officinale Seeds under SG and AG Conditions

Different concentrations of GA_3_ (0, 0.05, 0.1, 0.5, 1 μM) were added exogenously in medium to observe its effect on the SG and AG groups, respectively (Figure 3). These results showed that GA_3_ affected the establishment of the mycorrhizal relationship between fungus and seeds in a dose-dependent manner. Namely, seed germination did not significantly change in the low GA_3_ treatment concentration (0.05 μM), though germination was inhibited slightly by the middle GA_3_ concentration (0.1 μM), yet germination was completely inhibited when high concentrations of exogenous GA_3_ (0.5 μM, 1 μM) in SG compared to the control (SG without any GA_3_ treatment) (Figure 3A–T). During the 4 weeks after sowing seeds, their effective germination rate gradually decreased from 40% to 0%, while more exogenous GA_3_ was applied in SG. The resulting morphological characters examined under a light microscope showed the clear presence of pelotons in the embryo cell of SG seeds (Figure 3U–W). Seed germination was achieved to seedling differentiation stage (stage 4) under low GA_3_ treatment concentration (0.05 μM) in SG (Figure 3B,G,L). No fungal mycelium colonized the seed embryo in the 0.1 μM GA_3_ treatment in SG, indicating that exogenous GA_3_ at a high concentration probably inhibited the signal recognition that normally occurs between the fungus and the seed, leading to failed fungal colonization (Figure 3S–Y). Neither seed germination in 1/2 MS medium (without fungus) nor the mycelium growth of the fungus on PDA were inhibited or displayed conspicuous morphological changes at any exogenous GA_3_ concentration (Figure 3F–J,U1–Y1,A1–F1).

### 2.3. Identification of Hormone-Related Genes and Common Symbiosis Pathway-Related Genes and Their Expression Profiles in SG and AG of D. officinale Seeds

Based on previous RNA-Seq transcriptomic data of *D. officinale* seeds inoculated with *Tulasnella* sp., we screened upregulated genes involved in GA biosynthesis and signal transduction in SG, including those encoding ent–kaurene synthase (KS), ent–kaurene oxidase (KO), GA 20 oxidase (GA20ox), GA 3-beta dioxygenase (*GA3ox*), GA 2-oxidase (GA2ox), and DELLA protein (Table 2). In addition, the expression of predicted key enzyme genes involved in ABA biosynthesis (9-*cis*-epoxycarotenoid dioxygenase, NCED), metabolism (abscisic acid 8’-hydroxylase), and signal transduction (ABA responsive element binding factor) were also induced in SG of *D. officinale* seeds. Compared with AG, the genes related to IAA biosynthesis (YUCCA family monooxygenase and SAUR family protein) had upregulated expression in the SG stage (fold-change > 2.0 and false discovery rate (FDR) < 0.001) (Table 2).

The key genes involved in CSP were upregulated in SG in our transcriptomic database, such as calmodulin-like protein (Dendrobium_GLEAN_10048053) and the calcium-dependent protein kinase (Dendrobium_GLEAN_10016982) related to Ca^2+^ signal transduction. Putative mycorrhizal-induced genes, including those encoding bidirectional sugar transporter protein, chitinase, fatty acid desaturase, and aspartic proteinase were all significantly upregulated in SG compared to AG (Table 2).

Expression levels of putative genes involved in GAs biosynthesis such as *GA2ox (DoG2ox), GA3ox (DoGA3ox), GA3ox* (*DoGA3ox*) and encoding UDP-glucosyl transferase (*DoSGT*), G-box-binding factor (*DoGBF*), probable inactive receptor kinase (*DoIRK*), 9-*cis*-epoxycarotenoid dioxygenase (*DoNCED*), YUCCA family monooxygenase (*DOIPM*), SAUR family protein (*DoSAUR71*), calmodulin-like protein (*DoCML19*), calcium-dependent protein kinase (*DoCDPK26*), and nodulation signaling pathway protein (*DoNSP2-1, DoNSP2-2*) were validated by real-time quantitative PCR (qPCR); as were the mycorrhiza-induced genes encoding lysosomal pro-X carboxypeptidase (*DoPRCP*), hevamine A-like (*DoHAL*), glucan endo-1,3-beta-glucosidase (*DoGGLU*), bidirectional sugar transporter (*DoSWEET14*), beta-1,3-glucanase (*DoGLU*), and aspartic proteinase CDR1-like (*DoCDR1*) (Figure 4, Figure 5, Figure 6, Figure 7 and Figure 8). After seed germination (stage 2, stage 3, stage 4), all of these genes were usually highly expressed and upregulated in protocorm (stage 3) and seedling development stages (stage 4 and stage 5) in SG compared to AG of *D. officinale*.

### 2.4. Effect of Exogenous GA_3_ Treatment on Genes’ Expression

#### 2.4.1. Gene Expression Related to GAs Biosynthesis

After treatment with the exogenous GA_3_ concentrations, genes involved in GAs biosynthesis showed diverse expression patterns. We compared the differential expression of the above genes between asymbiotic and symbiotic conditions at a given treatment concentration GA_3_ treatment (Figure 4, Figure 5, Figure 6, Figure 7 and Figure 8). These results showed those genes related to the biosynthesis of GAs (*GA20ox, GA3ox*) were upregulated in seed germination (stage 2), protocorm formation (stage 3), and seedling (stage 4) in the SG, while *DoGA2ox* underwent significantly upregulated expression at the protocorm stage (stage 3) (Figure 4A). After applying exogenous GA_3_, the expression of *GA3ox* gene in SG was 10.19, 26.42, 74.74, 109.36, and 104.15 times that in AG at 0, 0.05 μM, 0.1 μM, 0.5 μM and 1 μM exogenous GA_3_ treatment concentrations, respectively (Figure 4B). In addition, the expression level of GA2ox, the key gene encoding gibberellin oxidase, which catalyzes the degradation of active GAs, was upregulated sharply at a higher GA_3_ treatment concentration (0.5 μM) in SG compared to AG (246.17 fold-change). This implied a crosstalk interaction between the biosynthesis and metabolism of GAs and mycorrhizal establishment.

#### 2.4.2. Gene Related to ABA Biosynthesis and Signaling Transduction

Treated with exogenous GA_3_ in the germination experiment, the genes involved in ABA biosynthesis and signaling transduction displayed diverse expression profiles (Figure 5A). For example, the gene DoNCED responsible for ABA biosynthesis was downregulated with a greater GA_3_ concentration, while the genes involved in the signal transduction of ABA (*DoSGT, DoIRK,* and *DoGBF*) were all upregulated, implying ABA metabolism has a very active response to a changed GA_3_ concentration (Figure 5B). The expression of genes participating in auxin biosynthesis also displayed a similar profile. Notably, DoIPM, a key gene that belongs to the YUCCA family was upregulated in SG compared to AG under the 0.5-μM GA_3_ treatment concentration (Figure 6).

#### 2.4.3. Expression Analysis of Putative Genes Involved in Mycorrhizal Symbiosis and Common Symbiosis Pathway

The putative symbiosis-specific expression genes, including *DoHAL, DoPRCP, DoGGLU, DoGLU, DoSWEET, DoCDR1, DoCDPK2,* and *DoNSP2* featured similar expression levels in SG after treatment with different concentrations of GA_3_. In the SG group with no exogenous GA_3_ treatment, the expression level of these genes increased substantially compared to AG, indicating the expression of these genes was induced by mycorrhizal fungi invasion. However, their expression underwent a similar change after imposing the exogenous GA_3_ treatment; namely, genes were at first highly expressed in 0.1 μM of exogenous GA_3_ but then suppressed as the GA_3_ treatment concentration increased (Figure 7 and Figure 8). The expression of *DoCDPK26* was not significantly changed in AG across the GA_3_ treatment concentrations but it was significantly and highly expressed in the 0.5-μM GA_3_ treatment in the SG group. Similarly, the gene *DoCML19* also was highly expressed in SG yet not significantly changed by exogenous GA_3_; this implied the expression of these two genes was induced by mycorrhizal fungi but each responded differently to the exogenous GA_3_ treatment.

## 3. Discussion

Symbiotic germination of orchid seeds involves the dual process of seed self-development and mutualistic interaction with their mycorrhizal fungi. Thus, the process is quite complex physiologically and ecologically. Orchid seeds are too tiny to perform genetic manipulations and this has inevitably limited the studies on their mechanisms of symbiotic germination, yet recent breakthroughs on arbuscular mycorrhiza have laid the foundation for investigating the SG of orchid seeds [16]. Recent studies show that the mycoheterotrophic symbiosis between orchids and mycorrhizal fungi possesses major components shared with mutualistic plant–mycorrhizal symbioses [17]. Many studies have revealed that plant hormones, especially gibberellins, are important factors affecting seed germination [10], and they are also critical for the establishment of mycorrhizal symbiosis [18,19]. In our study, the contents of five plant hormones (GA_3_, ABA, IAA, ZT, and JA) was determined at four different developmental stages of seed germination of the orchid *D. officinale*. Our results revealed that the mature and ungerminated seed have the highest ABA content (12.78 ng/g·FW) but this declined further along the seed germination process, and is consistent with two other studies [20,21]. A little GA_3_ was detected in the early germination stage of SG and AG group but the content is no significant difference between SG and AG group. Exogenous GA_3_ negligibly affected asymbiotic germination at all concentrations used in our study, a result supporting early statements by Arditti [6] that, in general, gibberellins appear to have no effect on germinating orchid embryos, in line as well with reported findings on asymbiotic germination testing by Hadley and Harvais [22]. However, exogenous gibberellins did significantly affect symbiotic germination in our study, implying its important role in mycorrhizal establishment. In addition, although the content of GA_3_ was similar between the symbiotic and asymbiotic groups, the ratio GA_3_/ABA changed faster at seedling development stage in SG, indicating fungal infection probably affected the balance of endogenous GAs and ABA. Previous results indicated the gibberellin/abscisic acid balance was capable of governing the seed germination of palm and maize plants [23,24]. In tomato, the level of GAs increases as a consequence of a symbiosis-induced mechanism requiring functional arbuscules that depends on a functional ABA pathway in mycorrhizal symbiosis during the establishment of arbuscular mycorrhiza [25]. Additionally, at least 130 forms of GAs have been identified to date yet only a handful of these (GA_1,_ GA_3,_ GA_4,_ GA_5,_ and GA_7_) are known to be biologically active [26]. Thus, in our next research project, we plan to quantify other active GAs molecules in *D. officinale* seeds.

The amount of IAA rose dramatically during the seed germination process, but especially during the seedling development stage of the SG group, indicating that IAA production was probably induced by mycorrhizal fungus in SG. Early research has shown that only traces of auxin occur in *Cypripedium* seed but none at all in *Dendrobium* seeds [6,27]; however, in our study, IAA was detected at relatively high content in the ungerminated stage and this content declined in the course of AG. The conflicting results are likely due the detection methods used. UHPLC is undoubtedly more sensitive for the quantification of trace amounts of plant hormones. Auxin is recognized as a secondary dormancy phytohormone, controlling seed dormancy and germination [28]. In addition, auxin metabolism and signaling also plays a crucial role in the modification of roots growth during their colonization by the ectomycorrhizal fungus *Laccria bicolor* [29]. Our result suggests IAA production was induced greatly during orchid mycorrhizal establishment, which provides a possible explanation for the faster differentiation of embryo when the seed of *D. officinale* was inoculated with the mycorrhizal fungus.

In this study, jasmonic acid (JA) content went undetected in ungerminated seeds and low JA (1.63 ng/g·FW) occurred in the free-living fungus, whereas the most JA was present in the early germination stage (stage 2) in AG (Table 1). JA is widely known to be involved in the response of plants to various stress factors, yet surprisingly little research has been carried out on JA’s roles in seed germination [30]. Work by Dave et al. [31] found no massive increase in their contents during seed maturation of *Arabidopsis*, suggesting their accumulation instead occurred during early seed development. A recent study reported crosstalk between JA and ABA contributed to modulating seed germination in bread wheat and *Arabidopsis* [32]. Evidently, more research is required to unravel the molecular mechanisms by which jasmonates regulate the germination of seeds.

Besides inducing plant hormone production, the mycorrhizal fungus itself also produces hormones and this may influence its plant partners in crucial ways. In our study, all five hormones were detected in the mycorrhizal fungus *Tulasnella* sp. As for the dynamic change of hormones in symbiotic germination group of *D. officinale* seeds, whether their production arose from mycorrhizal fungi or from host plant induced by fungus is still unclear and merits further exploration in the future.

Exogenous GA_3_ treatment had a dose-dependent effect on the SG of *D. officinale* seeds but did not significantly affect either the AG or free-living mycelium growth in the phenotype. Based on our initial results, we speculate the signal recognition between seed and their mycorrhizal fungi was probably impaired in some way by a higher concentration of GA_3_. We did not detect fungal invasion (colonization) of the seed embryo when using either 0.5 μM and 1.0 μM exogenous GA_3_. Under the microscope, we saw the seed embryo enlarged but no germination ensued at these high GA_3_ concentrations in the SG group (Figure 3S–Y). Furthermore, high concentrations of GA_3_ did not stop free-living mycelium from growing. A previous study has shown that GAs are phytohormones able to inhibit arbuscular mycorrhizal fungal infection by inhibiting arbuscular mycorrhizal hyphal entry into the host root where they suppressed the expression of Reduced Arbuscular Mycorrhization1 (RAM1) and RAM2 homologs that function in hyphal entry and arbuscule formation [19]. A similar scenario probably occurred in SG of *D. officinale* seeds.

Furthermore, after receiving the exogenous GA_3_, plant hormone-related genes such as biosynthesis and signal transduction of GA, ABA or IAA were characterized by a similar expression profile. Namely, sharply increasing expression in response to 0.5 μM exogenous GA_3_ followed by transcriptional downregulation; accordingly, we infer that exogenous GA_3_ disturbed the balance of endogenous hormones and crosstalk regulation occurred between GA, IAA, and ABA during the seed germination of *D. officinale* inoculated with the *Tulasnella* sp. fungus. Normally, genes involved in GA and IAA synthesis are highly expressed in SG, especially in the protocorm and seedling stages of orchids. The symbiosis between *Cymbidum goeringii* and a *Rhizoctonia*-like mycorrhizal fungus caused the release of hormones, which were able to promote the growth of *C. goeringii* seedlings [7]. Similarly, it has been demonstrated that auxin promotes *Arabidopsis* root growth by modulating its gibberellin response [33]. We plan to quantify the endogenous hormones to further confirm the relationship between hormone content and gene expression under an exogenous GA_3_ treatment during orchid seed germination.

Based on our previous RNA-seq and iTRAQ data, we found four proteins encoding genes involved in the common symbiotic signal pathway, including two genes function-annotated as nodulation signaling pathway protein (*DoNSP2-1* and *DoNSP2-2*) and two Ca^2+^ signal-related proteins, a calcium-dependent protein kinase (*DoCDPK26*), and a calmodulin-like protein (*DoCML19*). All these genes were highly expressed in SG but differed markedly. The Ca^2+^ signal is a universal second messenger, and increases in cytosolic Ca^2+^ concentration are among the earliest signaling events occurring in plants challenged with mutualistic partners or pathogens [34,35]. CDPK and CML are the two principal protein families of plant Ca^2+^ sensors [36]. The gene encoding CDPK was also identified from *D. officinale* roots infected by an orchid mycorrhizal fungus (*Mycena* sp.) by using the reverse transcription-polymerase chain reaction (RT-PCR) and rapid amplification of cDNA ends (RACE) [37]. In our study, the genes encoding CDPK (*DoCDPK26*) and CML (*DoCML19*) exhibited sharply higher expression levels in SG across the applied concentration gradient exogenous GA_3_, especially under 0.5 μM (for *DoCDPK26*) and 1.0 μM (for *DoCML19*), respectively. However, this expression of *DoCML19* was similar to SG lacking exogenous GA_3_ treatment, suggesting gene expression was induced by mycorrhizal fungi and only weakly related to exogenous GA_3._ Conversely, the gene *DoCDPK26* showed a significant different expression in SG group with versus without exogenous GA_3_ treatment, which implied that the CDPK and CML proteins probably participate in this plant–microbe interaction in different ways. Given the difficulty of genetically manipulating orchid seeds and orchid mycorrhizae, in our future research biochemical and physiological methods will be applied to confirm the mechanistic linkage between this plant hormone and Ca^2+^ signal during the SG of orchid seed, as well as changed Ca^2+^ concentrations across a gradient of exogenous GA_3_ during seed germination of *D. officinale*.

We also found that the expression of genes encoding probable mycorrhizal signaling pathway proteins (*DoNSP2-1* and *DoNSP2-2*) (function-annotated as nodulation signaling pathway proteins), both of which encode GARS-family transcriptional regulators, considerably increased under 0.5 μM exogenous GA_3_ treatment in SG compared to AG. This result suggests exogenous GA_3_ probably affected the mycorrhizal-specific gene expression by controlling the mycorrhizal-signaling pathway. Gibberellin’s ability to govern the nodulation signaling pathway in *Lotus japonicus* has been clarified by Maekawa et al. [38], who found that exogenous application of biologically active GA_3_ inhibited the formation of infection threads and nodules; hence they suspected GA halted the nodulation signaling pathway downstream of cytokinin, possibly at NSP2, which is required for Nod factor-dependent NIN expression. Whether a similar situation, in which GA inhibited the downstream gene expression of the mycorrhizal signaling pathway, occurs in orchid mycorrhiza needs to be confirmed (or not) in a co-culture system of orchid seedlings with its mycorrhizal fungi.

Several typical putative mycorrhizal-fungi-induced expression genes were identified in the SG of *D. officinale* seeds based on our transcriptomic data: *DoCDR1*, *DoGGLU*, *DoGLU*, *DoPRCP*, and *DoSWEET.* For these genes, hardly any expression happened in AG but they were highly expressed in specific ways among different development stages of SG for the *D. officinale* seeds. The gene *DoCDR1* encodes an aspartic protease. Studies have found that the aspartic protease gene in rice, OsCDR1, can induce defense responses in plants and increase plant resistance to bacterial and fungal diseases [39]. *DoCDR1* was also upregulated in different germination stages of SG in the absence of the GA_3_ treatment: low concentration of it did not cause this gene’s expression to change, but 0.1 μM endogenous GA_3_ treatment strongly elevated *DoCDR1*′s expression, suggesting that fungi induced it. Exogenous GA_3_ probably affected the expression level by interfering with the balance of endogenous hormones.

*DoGGLU* and *DoGLU* are two genes encoding β-1,3-glucanase, belonging to the pathogenesis-related proteins class that plays an important role in biotic and abiotic stress responses of plants [40]. It has been shown that colonization by mycorrhizal fungi in orchid root does not trigger strong plant defense responses in orchid mycorrhiza of *Serapias vomeracea* with *T. calospora*, given the nonstimulated expression of the plant’s defense genes [41]. However, our proteomic analysis showed that fungus invasion activated the plant defense reaction because genes encoding catalase isozyme, L-ascorbate peroxidase, and superoxide dismutase—all of which are enzymes involved in defense mechanisms—were upregulated during the SG of *D. officinale* seeds [7]. High expression levels of β-1,3-glucanase genes suggest the host plant probably produced an antifungal defense reaction, especially in the protocorm stage, via the lysis of pelotons so as to limit the extent of invasion during the SG of *D. officinale*. Finally, since the high GA_3_ treatment concentrations triggered the strong expression of *DoGGLU*, *DoGLU*, this indicated the genes respond to exogenous environment stress.

SWEET family sugar exporters in arbuscula mycorrhizal symbiosis in *Medicago truncatula* are known to play a vital role in the transport of glucose across the peri-arbuscular membrane to maintain arbuscular for a healthy mutually beneficial symbiosis [42]. Genes encoding SWEET family proteins are often expressed more in the symbiotic tissues of mycorrhizal protocorms of the orchid *S. vomeracea* with *T. calospora*. In our study, evidence for a similar phenomenon was found. Mycorrhiza-induced genes were specifically expressed in SG and its expression rose sharply under the 0.1-μM exogenous GA_3_ treatment; hence, these genes responded to a changed exogenous GA_3_ concentration during the SG of *D. officinale* seed. Therefore, we propose that GAs is involved in the crosstalk signal pathway between GAs biosynthesis and common symbiotic signal pathway during *D. officinale* seeds’ symbiotic germination and is thereby able to influence the expression of mycorrhizal-induced genes.

## 4. Materials and Methods

### 4.1. Plant Materials and Growing Conditions

Seeds of *D. officinale* were collected from a greenhouse in Jinhua County of Zhejiang Province, China, in November 2015. Mature capsules were surface sterilized, and their axenic seeds were stored at 4 °C in wax paper packets inside 1.5-mL sterilized tubes containing sterilized silica gel [7]. A mycorrhizal fungus that was a *Tulasnella* sp. (S6), isolated previously from root of *D. nobile*, was cultured in potato dextrose agar (PDA) medium. Symbiotic germination (SG) testing was carried out in oatmeal agar plates (OMA, 0.25% oat meal and 1% agar) and the asymbiotic germination (AG) testing was performed in 1/2 Murashige & Skoog (1/2 MS) medium without fungi, under a 12-h/12-h light/dark (L/D) cycle at 25 °C. In our previous work, we demonstrated this fungus is able to stimulate seed germination of *D. officinale* prior to AG, by reducing time to germination and increasing germination rate [7].

### 4.2. Determination of Endogenous Hormone during Seed Germination of D. officinale

Endogenous hormones, including gibberellic acid (GA_3_), abscisic acid (ABA), indole-3-acetic acid (IAA), *trans*-zeatin (ZT) and jasmonic acid (JA), were examined on a total of eight samples at three different developmental stages (stage 2, stage 3, and stage 4) of AG and SG, ungerminated seed, and free-living mycelium of fungus. Each sample consisted of three biological replicates. Standards of ABA, ZT, indole-3-aceticacid, GA_3_, and JA (Sigma, St. Louis, MO, USA) were used for the quantification of endogenous hormones. Hormone extraction and fractionation followed the description of Kojima et al. [43]. Briefly, 50–200 mg of fresh seeds or fungi were frozen in liquid nitrogen and homogenized with a lysis buffer (methanol:water:formic acid = 7.9:2:0.1) in a 2-mL microcentrifuge tube. The homogenate was kept at 4 °C for at least 15 h. After centrifugation at 10,000× *g* for 15 min, the ensuing supernatant was transferred to a new collection tube. The combined eluate was evaporated and then reconstituted with 1 mL of 1 M formic acid, and then the hormone-containing fraction was passed through an MAX column. Quantitative analysis was performed using ultra-high performance liquid chromatography (UHPLC, Agilent 1290 Infinity, Agilent, Santa Clara, CA, USA) coupled with tandem mass spectrometry (MS/MS, Agilent 6490 Triple Quadrupole, Agilent, Santa Clara, CA, USA). Automatic identification and integration of each MRM transition was done under default parameter settings in Masshunter software (Agilent, Santa Clara, CA, USA), but assisted with manual inspections. The mass spectral peak area of the analyte was taken as the ordinate, and a linear regression standard curve drawn with the concentration of the analyte as the abscissa, from which the regression equation was obtained. Then, the mass spectral peak area of the analyte of a given sample was substituted into the linear equation, to calculate the content of each endogenous hormone.

### 4.3. Exogenous GA_3_ Treatment on Symbiotic and Asymbiotic Germination of D. officinale

The concentration of exogenous GAs was selected in preliminary experiments, which spanned 0.05 μM to 1 μM, according to a previous study [19]. Seeds were sown in OMA medium (for SG with fungus) and ½ MS medium (for AG without fungus) with four concentrations of exogenous GA_3_ (0.05, 0.1, 0.5, and 1 μM; Beijing Solarbio Science & Technology Co., Ltd., Beijing, China). We designed four groups, including seeds in OMA medium (no fungus), fungus in OMA medium (no seeds) with different concentrations of exogenous GA_3_, and the normal AG group (seeds in 1/2MS without fungus) and SG group (seed in OMA medium with fungus) without exogenous GA_3_. Next, the petri dishes were incubated at 25 °C under a 12-h/12-h light/dark (L/D) cycle. Morphological changes during seed germination were observed daily, under a stereomicroscope and a Leica light microscope DM2500 (Leica Microsystems, Wetzlar, Germany).

### 4.4. Transcriptome Analysis by RNA-Seq

Transcriptome analysis of the eight samples was performed in our previous study; the samples corresponded to three different germination stages (stage 2, stage 3, and stage 4) of symbiotic and asymbiotic seeds, respectively, and to free-living mycelium of mycorrhizal fungus and ungerminated seeds [7]. The original transcriptomic data was deposited in the public NCBI and SRA database (accession No. PRJNA279934). Based on this transcriptomic data, we screened the putative genes involved in biosynthesis and signal transduction of plant hormones, the common symbiotic pathway, and specific gene expression in SG of *D. officinale.*

### 4.5. RNA Extraction and Quantitative Real-Time PCR

RNA extraction and quantitative RT-PCR was performed as described in our prior study [7]. Briefly, total RNA was extracted from 200 mg of seeds using the RNeasy Plant Mini Kit (Qiagen, CA, USA) and treated with an RNase-free DNase I digestion kit (Beijing Aidlab Biotech Company, Beijing, China) to remove any residual genomic DNA. Then 1 μg of RNA was reverse-transcribed to cDNA, using a reverse transcription system (Bio-Rad Laboratories, Inc., Richmond, CA, USA), and the cDNA equivalent to 25 ng of total RNA served as a template for each PCR reaction, carried out using SYBR Green supermix (Bio-Rad Laboratories, Inc., Richmond, CA, USA) with a final concentration of 1.6 mM of each primer. The primer sequences are listed in Appendix A. The qRT-PCR experiments were done using a SYBR Premix Ex TaqTM (Takara Biotechnology Co., Ltd., Dalian, China) on the LightCycler 480 machine (Roche Applied, Mannheim, Germany). PCR amplifications of three biological replicates were performed, which also included three distinct technical replicates. A no-template control (i.e., RNase-free water) was included for every qPCR run. Transcript abundance was normalized using the housekeeping gene *EF-1α* and a given gene’s expression level amount was calculated by the 2^−ΔΔC_T_^ method [44].

### 4.6. Data Analysis

The data of hormone content were analyzed with one-way ANOVA and the statistical analysis was performed using software SPSS 11.0. Data were presented as means ± SD from at least three independent experiments. *p* values < 0.05 were considered significant difference.

## 5. Conclusions

This study mainly explored the relationship between endogenous hormones and symbiotic germination of orchid *D. officinale* seeds and the effects upon seed germination from exogenous GA_3_. Endogenous hormonal change regulated the seed germination of *D. officinale* and mycorrhizal fungi invasion can greatly stimulate its host plant’s endogenous IAA accumulation. This could explain the faster differentiation of the embryo at the protocorm stage during symbiotic germination. Exogenous GA_3_ has a dose-dependent effect on the establishment of the mycorrhizal relationship between the fungus and seeds, such that a high concentration of GA_3_ probably acts upon the genes or proteins of the common symbiotic pathway, thereby inhibiting the recognition between orchid seeds and mycorrhizal fungi to further influence seed germination. Gene expression of the putative mycorrhizal-induced and symbiotic signal pathway responds to exogenous GA_3_ concentration change, implying GA_3_ contributes to the crosstalk between the hormone biosynthetic pathway and common symbiotic signal pathway. This study lays a foundation for the further exploration of seed germination, especially the symbiotic germination mechanism of orchid seeds.

## Figures and Tables

**Figure 1 ijms-21-06104-f001:**
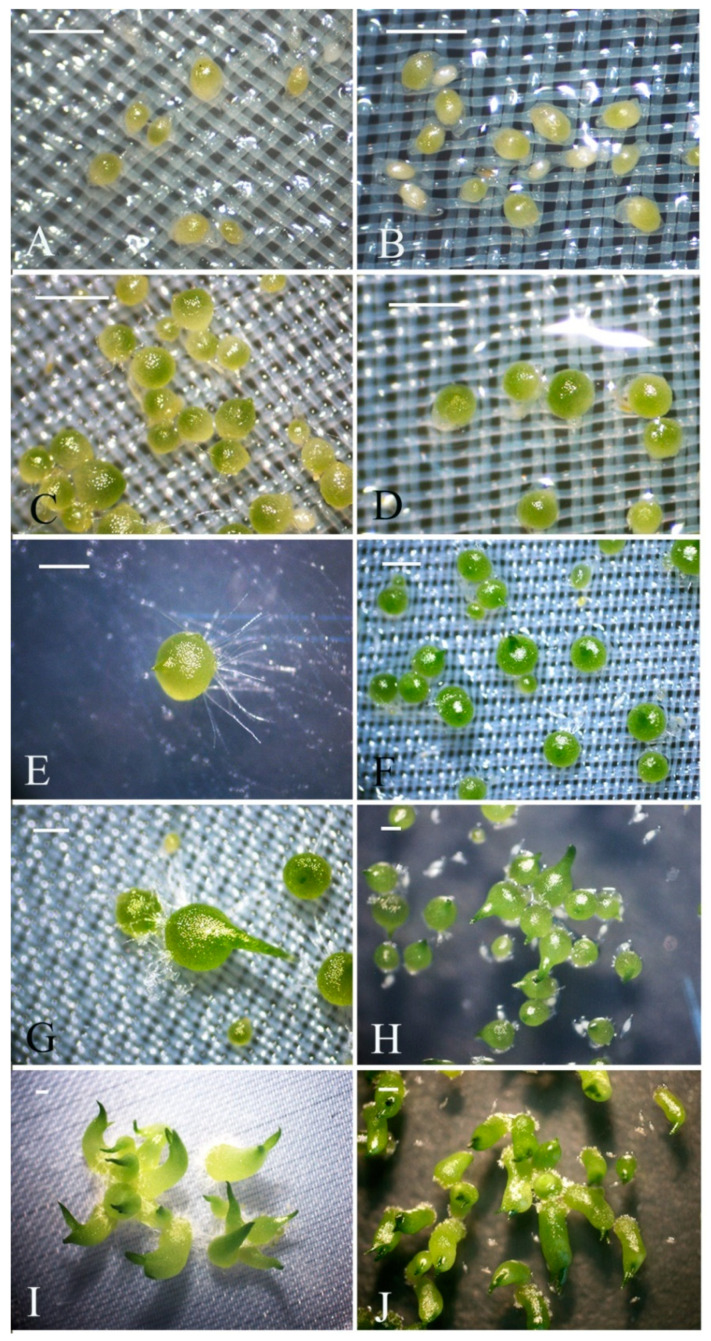
Morphological characters and seed developmental stages of *Dendrobium officinale. (***Left column**): symbiotic germination (**A**,**C**,**E**,**G**,**I**); (**right column**): asymbiotic germination (**B**,**D**,**F**,**H**,**J**). (**A**,**B**), stage 1: embryo swollen, turned light green, no germination; (**C**,**D**), stage 2: continued embryo enlargement, rupture of testa (germination); (**E**,**F**), stage 3: appearance of protomeristem (protocorm); (**G**–**H**), stage 4: emergence of first leaf (seedling); (**I**,**J**), stage 5: elongation of the first leaf. Scale bar = 0.5 mm.

**Figure 2 ijms-21-06104-f002:**
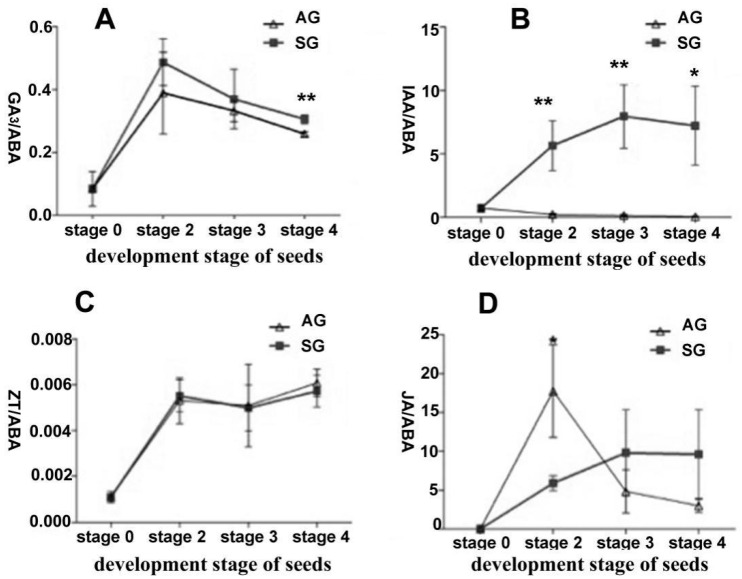
Endogenous hormone content change in different development stages during symbiotic and asymbiotic germination of *Dendrobium officinale* seeds. (**A**) GA_3_/ABA ratio in different stage between symbiotic and asymbiotic germination; (**B**) IAA/ABA ratio; (**C**) ZT/ABA ratio; (**D**) JA/ABA ratio. Mean and SE values were calculated from at least three replicates. Asterisks indicate significant differences in same development stage between asymbiotic and symbiotic germination according to the *t*-test (* *p* < 0.05 and ** *p* < 0.001). AG, asymbiotic germination; SG, symbiotic germination.

**Figure 3 ijms-21-06104-f003:**
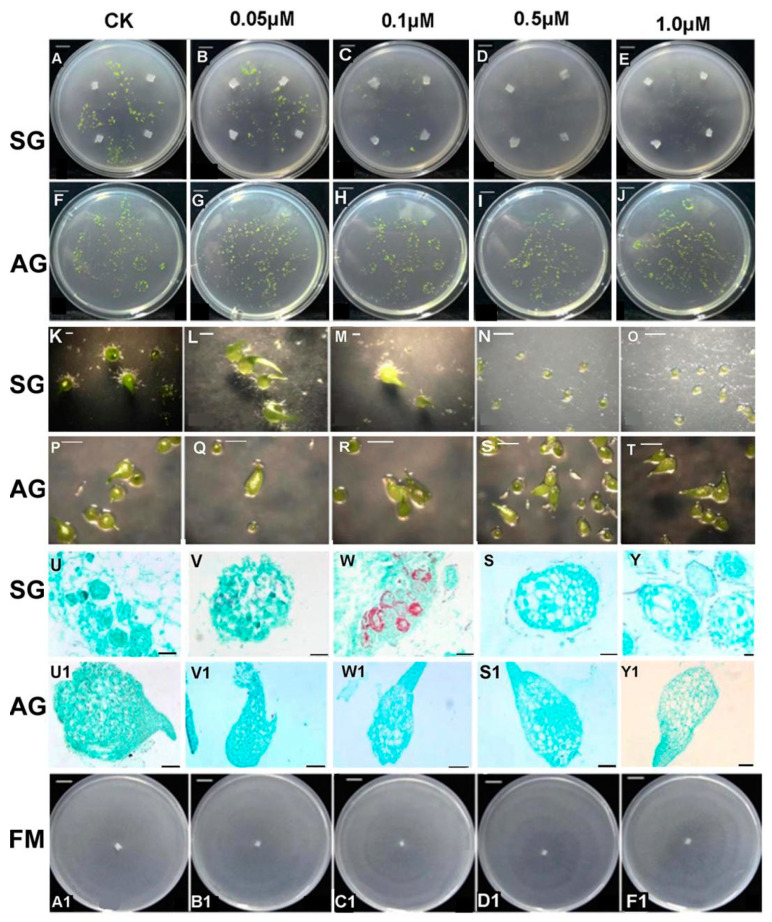
Effect of exogenous GA_3_ concentration on mycorrhizal fungi colonization in symbiotic and asymbiotic germination of *Dendrobium officinale* at 4 weeks after sowing seeds. (**A**–**J**). seed germination when inoculated with *Tulasnella* sp., or without the fungus, at different GA_3_ treatment concentrations. (**K**–**T**). morphological characters of symbiotic or asymbiotic germination at different GA_3_ treatment concentrations under a stereomicroscope; (**U**–**Y1**). morphological characters of symbiotic or asymbiotic germination at different GA_3_ treatment concentrations under a light microscope; (**A1**–**F1**). Colony of free-living mycelium of mycorrhizal fungus on PDA medium at different GA_3_ treatment concentrations. Scale bars: (**A**–**J**) = 1 cm; (**K**–**T**) = 1 mm; (**J**–**W**), (**U1**–**Y1**) = 10 μm; (**S**–**Y**) = 5 μm; (**A1**–**F1**) = 1 cm; SG, symbiotic germination; AG, asymbiotic germination.

**Figure 4 ijms-21-06104-f004:**
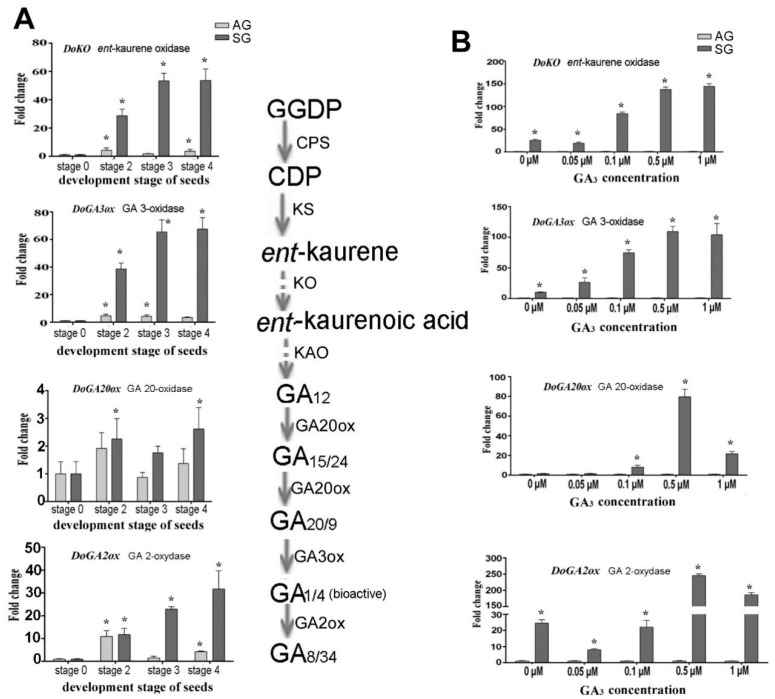
Expression levels of genes related to GA biosynthesis during symbiotic germination of *Dendrobium officinale* for quantitative qPCR analysis. (**A**). Genes’ expression at different development stages between asymbiotic and symbiotic germination; (**B**). Genes’ expression affected by GA concentrations at 4 weeks after sowing seeds. Note the fold-change values are relative to asymbiotic germination. PCR amplifications were performed for three biological replicates and two distinct technical replicates for each sample. Expression levels were calculated by the 2^−ΔΔC_T_^ method normalized against the expression of *EF1-α**,* using the expression level of ungerminated seed (stage 0) (**A**) or asymbiotic germination (**B**) as control and the fold change > 2.0 was marked significant differential expression (*). AG, asymbiotic germination; SG, symbiotic germination.

**Figure 5 ijms-21-06104-f005:**
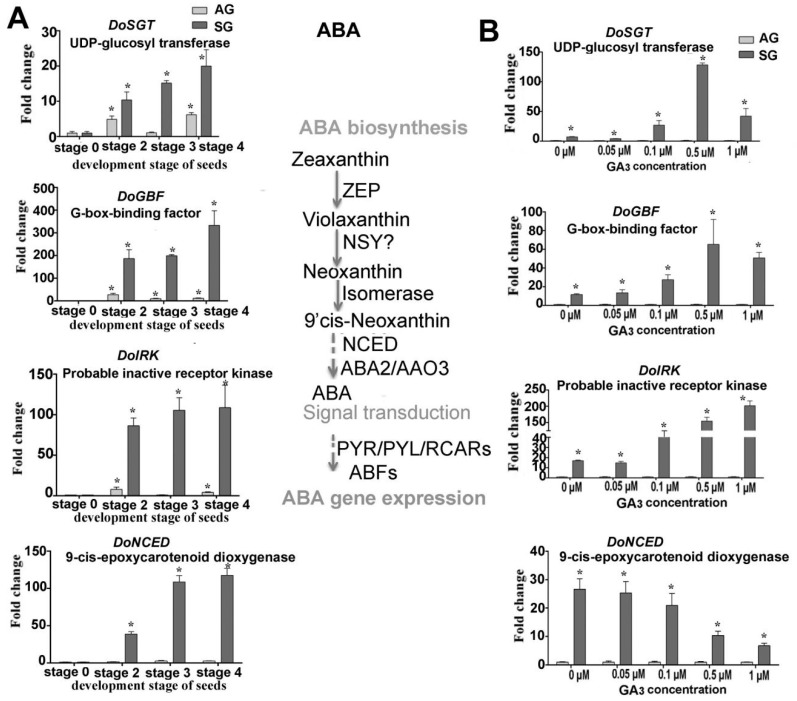
Expression levels of genes related to ABA biosynthesis during symbiotic germination of *Dendrobium officinale* for quantitative qPCR analysis. (**A**). Genes’ expression at different development stages between asymbiotic and symbiotic germination (no GA_3_ treatment); (**B**). Genes expression’ affected by GA_3_ concentrations at 4 weeks after sowing seeds. Expression levels were calculated by the 2^−ΔΔC_T_^ method normalized against the expression of *EF1-α**,* using the expression level of ungerminated seed (stage 0) (**A**) or asymbiotic germination (**B**) as control and the fold change > 2.0 was marked significant differential expression (*). AG, asymbiotic germination; SG, symbiotic germination.

**Figure 6 ijms-21-06104-f006:**
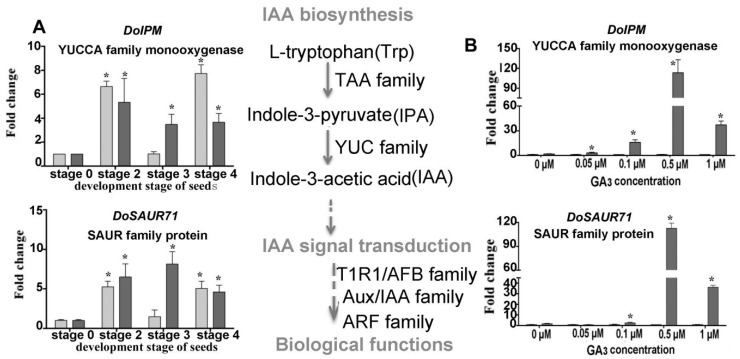
Expression levels of genes related to IAA biosynthesis during symbiotic germination of *Dendrobium officinale* for quantitative qPCR analysis. (**A**). Genes’ expression at different development stages between asymbiotic and symbiotic germination (no GA_3_ treatment); (**B**). Genes’ expression was affected by GA_3_ concentrations at 4 weeks after sowing seeds. Expression levels were calculated by the 2^−ΔΔC_T_^ method normalized against the expression of *EF1-α,* using the expression level of ungerminated seed (stage 0) (**A**) or asymbiotic germination (**B**) as control and the fold change > 2.0 was marked significant differential expression (*). AG, asymbiotic germination; SG, symbiotic germination.

**Figure 7 ijms-21-06104-f007:**
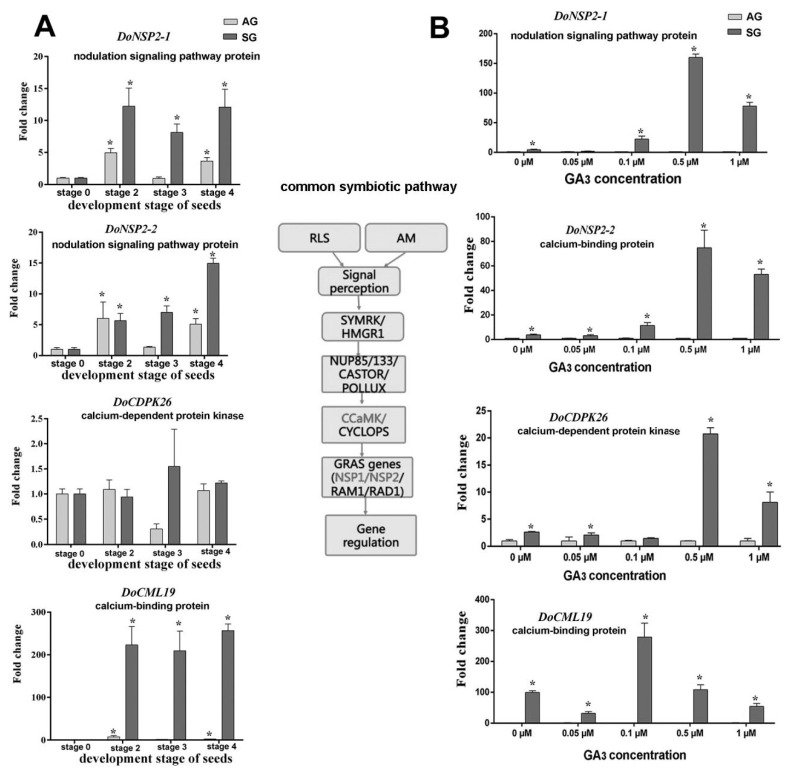
Expression levels of genes related to common symbiosis pathway during symbiotic germination of *Dendrobium officinale* for quantitative qPCR analysis. (**A**). Genes’ expression at different development stages between asymbiotic and symbiotic germination (no GA_3_ treatment); (**B**). Genes’ expression was affected by GA_3_ concentrations at 4 weeks after sowing seeds. Expression levels were calculated by the 2^−ΔΔC_T_^ method normalized against the expression of *EF1-α,* using the expression level of ungerminated seed (stage 0) (**A**) or asymbiotic germination (**B**) as control and the fold change > 2.0 was marked significant differential expression (*). AG, asymbiotic germination; SG, symbiotic germination.

**Figure 8 ijms-21-06104-f008:**
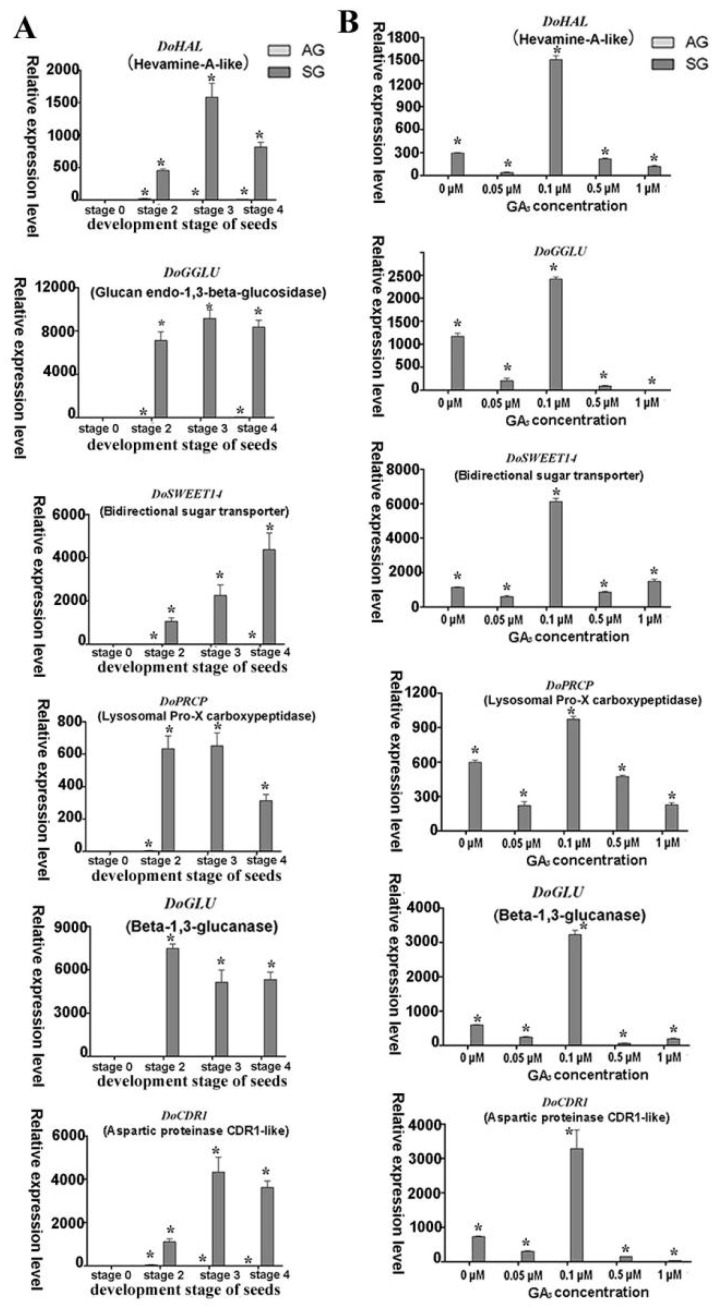
Expression levels of putative mycorrhiza-induced genes involved in orchid symbiotic germination of *Dendrobium officinale* for quantitative qPCR analysis. (**A**). Genes’ expression at different development stages between asymbiotic and symbiotic germination (no GA_3_ treatment); (**B**). Genes’ expression was affected by GA_3_ concentrations at 4 weeks after sowing seeds. Expression levels were calculated by the 2^−ΔΔC_T_^ method normalized against the expression of *EF1-α,* using the expression level of ungerminated seed (stage 0) (**A**) or asymbiotic germination (**B**) as control and the fold change > 2.0 was marked significant differential expression (*). AG, asymbiotic germination; SG, symbiotic germination.

**Table 1 ijms-21-06104-t001:** The content of five kinds of endogenous plant hormones during the seed germination of *Dendrobium officinale* (*n* = 3). Abbreviation: FW: fresh weight; U0: ungerminated seeds; SG: symbiotic germination; AG: asymbiotic germination; 2, 3, 4 means stage 2, stage 3 and stage 4 during the germination of *D. officinale* seed, respectively; S6 means the free-living mycelium on PDA of our mycorrhizal fungus; GA_3_: gibberellic acids; ABA: abscisic acids. IAA: indole-3-acetic acid; ZT: *trans*-zeatin; JA: jasmonic acid.

Sample	ABA (ng/g·FW)	GA_3_ (ng/g·FW)	IAA (ng/g·FW)	ZT (ng/g·FW)	JA (ng/g·FW)
U0	12.78 ± 2.87 b	0.99 ± 0.46 abc	9.00 ± 1.2 ab	0.014 ± 0.00 ab	0 a
SG2	2.89 ± 0.40 a	1.40 ± 0.27 cd	16.52 ± 7.13 b	0.016 ± 0.00 bc	16.85 ± 1.37 abc
SG3	3.21 ± 0.50 a	1.19 ± 0.35 bc	25.91 ± 9.66 c	0.052 ± 0.00 bc	31.76 ± 18.57 c
SG4	2.15 ± 0.17 a	0.66 ± 0.08 ab	15.70 ± 7.51 b	0.012 ± 0.00 ab	21.16 ± 13.95 bc
AG2	5.06 ± 1.00 a	1.90 ± 0.37 d	1.22 ± 0.40 a	0.026 ± 0.00 d	85.95 ± 14.63 d
AG3	2.85 ± 0.75 a	0.95 ± 0.31 abc	0.48 ± 0.36 a	0.014 ± 0.00 ab	12.96 ± 7.37 abc
AG4	3.35 ± 1.30 a	0.87 ± 0.36 abc	0.19 ± 0.14 a	0.021 ± 0.01 cd	9.38 ± 0.89 ab
S6	4.29 ± 2.79 a	0.44 ± 0.05 a	4.09 ± 2.19 a	0.0075 ± 0.00 a	1.63 ± 0.07 ab

Note: Different letters a, b, c, d represent significant difference (*p* < 0.05) of content of phytohormone in different development stage of *D. officinale* seeds. The analysis was performed using the Duncan method in SPSS 17.0 software.

**Table 2 ijms-21-06104-t002:** Putative mycorrhizal induced genes, GA biosynthesis and other hormone gene homologs induced during symbiotic germination of *D. officinale* seeds. Abbreviation: FPKM: fragments per kilobase per million and the value represents the differential expression level; FDR: false discovery rate; A1, A2, A3 represent asymbiotic germination stage 2, stage 3, and stage 4 respectively; S1, S2, S3 represent symbiotic germination stage 2, stage 3, and stage 4, respectively. * gene expression were validated by qPCR.

Pathway	KEGG Description	Gene ID	FPKM											
			A1	S1	A2	S2	A3	S3	log2(S1/A1)	FDR	log2(S2/A2)	FDR	log2(S3/A3)	FDR
**GA biosynthesis and metabolism**	**ent-kaurene synthase [EC:4.2.3.19]** (*KS*)	Dendrobium_GLEAN_10038616	0.00	9.32	0.00	3.30	0.67	6.56	13.19	1.06 × 10^−11^	11.69	5.00 × 10^−3^	3.29	3.68 × 10^−5^
	**ent-kaurene oxidase [EC:1.14.13.78]** (*KO*)	Dendrobium_GLEAN_10138923 *	0.60	30.24	0.69	40.62	0.45	15.44	5.66	1.22 × 10^−49^	5.88	4.93 × 10^−63^	5.10	4.57 × 10^−24^
		Dendrobium_GLEAN_10138922	2.02	76.16	1.15	64.58	0.96	36.56	5.24	1.08 × 10^−38^	5.81	6.06 × 10^−31^	5.25	1.23 × 10^−16^
	gibberellin 2-oxidase [EC:1.14.11.13] (*GA2ox*)	Dendrobium_GLEAN_10025219 *	1.72	97.72	0.00	119.45	0.28	51.62	5.83	4.63 × 10^−87^	16.87	2.52 × 10^−109^	7.53	3.21 × 10^−47^
		Dendrobium_GLEAN_10027180	3.64	74.66	7.80	30.53	1.56	12.74	4.36	5.08 × 10^−63^	1.97	3.56 × 10^−11^	3.03	7.02 × 10^−8^
		Dendrobium_GLEAN_10062493	23.77	78.49	20.87	54.56	0.00	31.03	1.72	2.15 × 10^−22^	1.39	6.37 × 10^−11^	14.92	1.11 × 10^−27^
		Dendrobium_GLEAN_10062492	7.36	51.88	15.18	51.46	0.00	15.66	2.82	2.88 × 10^−28^	1.76	4.23 × 10^−15^	13.93	4.75 × 10^−15^
		Dendrobium_GLEAN_10040612	2.73	32.22	6.21	17.34	0.00	10.67	3.56	1.90 × 10^−12^	1.48	6.00 × 10^−3^	13.38	2.23 × 10^−5^
	**gibberellin 3-beta-dioxygenase [EC:1.14.11.15]** (*GA3ox*)	Dendrobium_GLEAN_10043501 *	9.70	48.02	4.69	46.72	1.95	28.86	2.31	1.32 × 10^−30^	3.32	8.54 × 10^−41^	3.89	1.54 × 10^−30^
	**gibberellin 20-oxidase [EC:1.14.11.12]** (*GA20 ox*)	Dendrobium_GLEAN_10090677	20.94	31.84	3.19	32.77	11.55	33.28	0.60	1.3 × 10^−2^	3.36	3.27 × 10^−16^	1.53	3.71 × 10^−6^
		Dendrobium_GLEAN_10048964 *	19.32	67.28	7.35	40.77	7.45	23.72	1.80	6.2 × 10^−22^	2.47	3.29 × 10^−18^	1.67	1.83 × 10^−6^
		Dendrobium_GLEAN_10048963	11.72	39.81	3.22	17.42	2.18	9.39	1.76	1.48 × 10^−12^	2.44	8.50 × 10^−8^	2.11	6.00 × 10^−4^
	momilactone-A synthase [EC:1.1.1.295]	Dendrobium_GLEAN_10065156	11.57	36.43	10.14	21.53	4.64	19.74	1.65	5.60 × 10^−13^	1.09	2.0 × 10^−4^	2.09	9.71 × 10^−9^
	isoprene synthase [EC:4.2.3.27]	Dendrobium_GLEAN_10097477	8.69	20.33	5.75	51.87	4.84	21.22	1.23	1.0 × 10^−4^	3.17	2.17 × 10^−29^	2.13	3.21 × 10^−8^
**GA signal transduction**	2-hydroxyisoflavanone dehydratase [EC:4.2.1.105]	Dendrobium_GLEAN_10030908	0.17	1.15	0.20	12.38	0.00	15.51	2.76	9.3 × 10^−2^	5.95	4.03 × 10^−17^	13.92	6.73 × 10^−25^
	DELLA protein	Dendrobium_GLEAN_10018544	0.00	13.28	0.41	21.73	0.24	36.19	13.70	2.27 × 10^−30^	5.73	2.14 × 10^−41^	7.24	1.76 × 10^−79^
		Dendrobium_GLEAN_10067459	14.06	120.53	1.54	74.55	2.13	43.34	3.10	2.79 × 10^−74^	5.60	7.66 × 10^−64^	4.35	1.77 × 10^−33^
		Dendrobium_GLEAN_10068560	11.66	106.49	12.73	41.88	6.48	43.20	3.19	1.0 × 10^−136^	1.72	8.15 × 10^−25^	2.74	4.32 × 10^−47^
		Dendrobium_GLEAN_10060070	34.54	97.03	21.39	93.50	26.06	141.08	1.49	1.58 × 10^−45^	2.13	5.49 × 10^−67^	2.44	8.15 × 10^−123^
		Dendrobium_GLEAN_10024051 *	0.00	0.59	0.00	2.23	1.05	5.65	9.20	1.15 × 10^−1^	11.12	3.0 × 10^−4^	2.43	1.11 × 10^−5^
**ABA biosynthesis and metabolism**	zeta-carotene desaturase [EC:1.3.5.8]	Dendrobium_GLEAN_10021410	0.00	518.36	0.19	474.48	0.00	80.22	18.98	0	11.29	0	16.29	6.05 × 10^−137^
	**9-*cis*-epoxycarotenoid dioxygenase [EC:1.13.11.51]** (*NCED*)	Dendrobium_GLEAN_10070249 *	1.74	10.23	0.50	43.83	1.17	24.04	2.56	6.99 × 10^−10^	6.45	2.31 × 10^−72^	4.36	3.32 × 10^−35^
	abscisic acid 8’-hydroxylase [EC:1.14.13.93]	Dendrobium_GLEAN_10055771	27.91	129.63	54.03	68.91	34.40	109.62	2.22	4.28 × 10^−115^	0.35	6.0 × 10^−4^	1.67	1.59 × 10^−64^
	momilactone-A synthase [EC:1.1.1.295]	Dendrobium_GLEAN_10065156	11.57	36.43	10.14	21.53	4.64	19.74	1.65	5.60 × 10^−13^	1.09	2.0 × 10^−4^	2.09	9.71 × 10^−9^
**ABA signal and transduction**	**ABA responsive element binding factor** (*ABF*)	Dendrobium_GLEAN_10081660 *	11.98	141.30	12.06	88.75	5.13	54.98	3.56	1.99 × 10^−112^	2.88	4.30 × 10^−54^	3.42	2.13 × 10^−40^
	abscisic acid receptor PYR/PYL family	Dendrobium_GLEAN_10090645	19.08	66.48	14.51	96.18	22.40	100.66	1.80	2.20 × 10^−15^	2.73	6.49 × 10^−32^	2.17	4.12 × 10^−23^
		Dendrobium_GLEAN_10028158	28.88	74.18	36.12	106.46	20.79	66.96	1.36	9.24 × 10^−17^	1.56	3.17 × 10^−26^	1.69	3.42 × 10^−17^
	**ubiquitin-conjugating enzyme E2 H [EC:6.3.2.19]** (*UBE2N*)	Dendrobium_GLEAN_10040978 *	0.20	100.23	0.07	64.50	0.00	32.99	8.97	1.12 × 10^−269^	9.85	6.43 × 10^−163^	15.01	1.04 × 10^−89^
**IAA biosynthesis and metabolism**	indoleacetaldoxime dehydratase [EC:4.99.1.6]	Dendrobium_GLEAN_10124247	6.48	12.52	5.33	19.60	4.10	31.70	0.95	1.00 × 10^−3^	1.88	6.42 × 10^−12^	2.95	2.10 × 10^−33^
	**YUCCA family monooxygenase [EC:1.14.13.8]** (*YUCCA*)	Dendrobium_GLEAN_10119687	0.14	38.55	0.00	49.20	0.28	24.59	8.11	2.22 × 10^−74^	15.59	9.99 × 10^−91^	6.46	5.28 × 10^−45^
		Dendrobium_GLEAN_10046914	0.99	37.15	0.00	51.97	0.73	37.02	5.23	3.77 × 10^−21^	15.67	2.55 × 10^−30^	5.66	2.96 × 10^−20^
		Dendrobium_GLEAN_10072646	0.47	7.76	2.85	15.23	1.32	21.73	4.05	1.47 × 10^−3^	2.42	4.14 × 10^−15^	4.04	1.04 × 10^−36^
		Dendrobium_GLEAN_10012601 *	0.00	29.08	0.00	39.65	0.84	20.41	14.83	2.90 × 10^−35^	15.28	2.46 × 10^−44^	4.60	6.04 × 10^−20^
		Dendrobium_GLEAN_10061932	4.93	20.80	8.44	42.64	3.07	22.84	2.08	2.89 × 10^−15^	2.34	4.02 × 10^−33^	2.90	1.34 × 10^−23^
		Dendrobium_GLEAN_10046916	4.01	30.78	1.65	44.97	0.34	33.09	2.94	3.33 × 10^−33^	4.77	9.35 × 10^−66^	6.60	1.00 × 10^−59^
**IAA signal transduction**	SAUR family protein	Dendrobium_GLEAN_10110599	1.27	46.87	0.48	27.33	0.41	10.16	5.21	6.32 × 10^−28^	5.83	1.38 × 10^−15^	4.63	1.57 × 10^−5^
		Dendrobium_GLEAN_10011274	2.42	65.45	0.69	24.78	0.00	9.10	4.76	3.59 × 10^−27^	5.17	8.22 × 10^−10^	13.15	3.00 × 10^−4^
		Dendrobium_GLEAN_10033477	0.00	137.57	0.00	44.21	0.00	10.04	17.07	1.59 × 10^−71^	15.43	1.72 × 10^−20^	13.29	8.65 × 10^−5^
		Dendrobium_GLEAN_10011273	4.55	61.11	0.52	22.76	0.00	18.76	3.75	6.23 × 10^−28^	5.45	6.40 × 10^−12^	14.20	4.45 × 10^−11^
		Dendrobium_GLEAN_10033975*	1.76	27.04	2.67	29.54	1.64	34.50	3.94	1.14 × 10^−10^	3.47	1.70 × 10^−9^	4.39	3.53 × 10^−12^
		Dendrobium_GLEAN_10001422	1.25	17.82	0.00	8.76	1.74	13.04	3.83	9.05 × 10^−7^	13.10	5.00 × 10^−4^	2.91	1.00 × 10^−3^
		Dendrobium_GLEAN_10116922	0.00	4.71	0.00	1.44	0.29	3.71	12.20	9.15 × 10^−5^	0.16	7.2 × 10^−1^	0.23	7.99 × 10^−1^
		Dendrobium_GLEAN_10075175	0.00	6.74	0.00	0.63	0.00	0.00	12.72	4.00 × 10^−4^	9.30	5.8 × 10^−1^	-	-
**Ca^2+^ signial related**	extracellular signal-regulated kinase [EC:2.7.11.24]	Dendrobium_GLEAN_10064705	0.00	16.86	0.40	5.95	0.00	16.86	14.04	6.55 × 10^−13^	3.89	1.00 × 10^−3^	14.04	6.55 × 10^−13^
	calcium-binding protein CML	Dendrobium_GLEAN_10048053 *	0.45	13.61	4.33	86.28	0.45	13.61	4.92	1.18 × 10^−6^	4.32	8.12 × 10^−38^	4.92	1.18 × 10^−6^
		Dendrobium_GLEAN_10140021	0.80	13.82	0.00	53.38	0.80	13.82	4.11	6.54 × 10^−7^	15.70	8.13 × 10^−34^	4.11	6.54 × 10^−7^
	calcium-dependent protein kinase [EC:2.7.11.1]	Dendrobium_GLEAN_10016982 *	0.00	10.17	0.62	7.70	0.00	10.17	13.31	1.78 × 10^−10^	3.63	9.86 × 10^−6^	13.31	1.78 × 10^−10^
		Dendrobium_GLEAN_10079575	0.80	17.06	3.19	17.30	0.80	17.06	4.41	1.86 × 10^−34^	2.44	3.90 × 10^−19^	4.41	1.86 × 10^−34^
		Dendrobium_GLEAN_10071601	1.40	14.50	2.14	23.68	1.40	14.50	3.37	1.69 × 10^−16^	3.47	3.14 × 10^−40^	3.37	1.69 × 10^−26^
		Dendrobium_GLEAN_10022554	2.48	11.31	8.94	41.67	2.48	11.31	2.19	3.00 × 10^−2^	2.22	2.41 × 10^−11^	2.19	3.00 × 10^−3^
		Dendrobium_GLEAN_10064211	6.40	61.24	1.65	45.27	6.40	61.24	3.26	8.29 × 10^−71^	4.78	2.30 × 10^−66^	3.26	8.29 × 10^−71^
		Dendrobium_GLEAN_10022555	11.00	94.82	31.00	171.02	11.00	94.82	3.11	1.12 × 10^−90^	2.46	2.24 × 10^−122^	3.11	1.12 × 10^−90^
**The common symbiotic pathway**	nodulation-signaling pathway 2 protein	PEQU_11738-D2 *	0.87	27.28	1.40	18.44	0.00	10.96	4.97	1.85 × 10^−36^	3.72	3.32 × 10^−19^	13.42	3.44 × 10^−17^
		Dendrobium_GLEAN_10030409 *	2.24	138.59	2.42	80.85	1.55	68.84	5.95	1.32 × 10^−295^	5.06	4.49 × 10^−149^	5.47	2.04 × 10^−141^
**putative mycorrhiza-induced genes**	aspartic proteinase CDR1-like [EC:2.7.1.-]	Dendrobium_GLEAN_10074206	0.00	727.07	0.00	629.87	0.00	435.89	19.47	0	19.26	0	18.73	0
	glucan endo-1,3-beta-glucosidase-like isoform X1	Dendrobium_GLEAN_10113668 *	0.00	1685.91	0.00	795.80	0.00	379.47	20.69	0	19.60	0	18.53	0
	Non-specific lipid-transfer protein	Dendrobium_GLEAN_10036826	0.00	1460.16	0.70	1564.23	0.00	275.31	20.48	0	123.78	0	18.07	5.83 × 10^−121^
	bidirectional sugar transporter SWEET14-like	Dendrobium_GLEAN_10125587 *	0.00	216.32	0.00	228.84	0.00	239.67	17.72	1.96 × 10^−205^	17.80	1.46 × 10^−199^	17.87	4.20 × 10^−218^
	Aspartic proteinase nepenthesin-1 precursor	Dendrobium_GLEAN_10098789	0.13	739.91	0.00	439.57	0.00	217.37	12.47	0	18.75	0	17.73	0
	Subtilisin-like serine endopeptidase family protein	Dendrobium_GLEAN_10135421	0.00	336.63	0.00	369.14	0.00	211.92	18.36	0	18.49	0	17.69	0
	non-specific lipid-transfer protein-like protein	Dendrobium_GLEAN_10080723	-	-	0.48	583.44	0.00	193.03	-	-	105.01	0	17.56	4.93 × 10^−125^
	chitinase [EC:3.2.1.14]	Dendrobium_GLEAN_10053378	0.00	124.35	0.00	198.49	0.00	183.43	16.92	9.45 × 10^−96^	17.60	8.18 × 10^−140^	17.48	7.11 × 10^−133^
		Dendrobium_GLEAN_10042237 *	0.00	106.39	0.00	167.62	0.00	153.94	16.70	9.42 × 10^−151^	17.35	1.86 × 10^−22^	17.23	2.75 × 10^−215^
	fatty acid desaturase [EC:3.1.4.4]	Dendrobium_GLEAN_10098792 *	0.00	493.08	0.00	423.04	0.00	147.60	18.91	0	18.69	0	17.17	1.83 × 10^−281^
	beta-1,3-glucanase	Dendrobium_GLEAN_10033071 *	0.36	783.01	0.20	383.72	0.00	136.56	11.09	0	118.94	0	17.06	8.82 × 10^−213^
	glucan endo-1,3-beta-glucosidase-like isoform X1	Dendrobium_GLEAN_10050850	0.00	391.39	0.00	273.17	0.00	122.44	18.58	0	18.06	0	16.90	2.09 × 10^−197^
	mannose-specific lectin 3-like	Dendrobium_GLEAN_10079890	1.19	2499.75	0.27	1267.07	0.24	1020.79	11.04	0	148.75	0	12.05	0
	lysosomal Pro-X carboxypeptidase-like [EC:3.4.16.2]	Dendrobium_GLEAN_10008416*	0.11	1471.74	0.00	1564.87	0.23	765.35	13.71	0	20.58	0	11.70	0
	mannose-specific lectin-like	Dendrobium_GLEAN_10005758	2.20	2498.33	1.01	1576.93	0.84	989.65	10.15	0	112.54	0	10.20	0
	subtilisin-like protease SDD1 [EC:3.4.21.112]	Dendrobium_GLEAN_10069506	1.14	1423.95	0.35	1729.18	0.85	950.92	10.29	0	150.56	0	10.13	0

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
