# Peer review of "Symbiotic and Asymbiotic Germination of Dendrobium officinale (Orchidaceae) Respond Differently to Exogenous Gibberellins"

_ijms, 2020, doi:10.3390/ijms21176104_

Round 1

Reviewer 1 Report

Seed germination of orchids is a complex process due to the specific properties of the dust like small seeds. The work presented endeavored at unraveling the molecular biology of hormonal control of the process. The article presents a rich set of data on morphological observations and molecular analyses of the control by phytohormones, especially gibberellins, of various stages of germination in relation to the paramount role of mycorrhiza. It advances our understanding and opens ways for future investigations. The introduction and discussion are informative and put observations well into perspective. Publication is suggested after minor revision considering the presentation.
L98 delete “gradually”, it is much lower indeed but gradually lower is not seen
L105 but in the actual table there are different values
L110 in the actual table it is not 4.28 but 4.29
Figures 1, 3: the panels are very small and the observations described are hard to see
L204 what is SF?
L373 onwards and elsewhere: Make it a bit clearer what you mean here by nodulation. It is an analogy. Orchid roots do not nodulate.
The language is good to read and clear. In some cases, some polishing could improve it. Some examples are given below, although they are not completely covering the polishing needed and a general treatment of the text would be nice.
L36 … minerals nutrients …
L50 … of the embryo …
L77 replace comma by a second hyphen … symbiosis – transducing ….
L81 an orchid
L85 hormones level
L116 … plant hormones during …
L143 “can up” not clear language
L174 and page 8: Ca2
L197 ??? … biosynthesis: After …
L270 … on their mechanisms of …
L278 … revealed that the mature …
L280-282 Sentence is not clear, grammar is insufficient.
L324 … by fungus is still unclear …
L333 … that GAs are …
L335 … where they suppressed …
L361 the genes encoding …
L373 … found that the expression …
L395 delete the second “genes”
L397 … by mycorrhizal fungi…
L408 arbuscular
L432 gibberellic acid
L485 … analysis was performed …
L495 … concentration of GA3 ….

Author Response

Respond to reviewers’ comments

August. 18, 2020

Manuscript Number: ijms-908896
Title: Symbiotic and Asymbiotic Germination of Dendrobium officinale (Orchidaceae) Respond Differently to Exogenous Gibberellins

According to editor and reviewers’ helpful suggestions, we have revised our manuscript point by point and uploaded the revision file in the submission system. Our specific responses to the comments are listed below. Original comments raised by the editor and reviewers are in italics, and our responses are in regular font.

Reviewer I

Seed germination of orchids is a complex process due to the specific properties of the dust like small seeds. The work presented endeavored at unraveling the molecular biology of hormonal control of the process. The article presents a rich set of data on morphological observations and molecular analyses of the control by phytohormones, especially gibberellins, of various stages of germination in relation to the paramount role of mycorrhiza. It advances our understanding and opens ways for future investigations. The introduction and discussion are informative and put observations well into perspective. Publication is suggested after minor revision considering the presentation.

Thanks for your helpful suggestions and encouragement.

  1. L98 delete “gradually”, it is much lower indeed but gradually lower is not seen

Thanks, we deleted the word ”gradually”. thanks for your help

  1. L105 but in the actual table there are different values

L110 in the actual table it is not 4.28 but 4.29

Thanks for your correction. We are sorry for our careless. We checked carefully the original data and corrected it. “Additionally, minute amounts of ZT (0.0075~0.014 ng/g.FW) were detected in both free-living mycelium of fungus and ungerminated seeds. For ungerminated seeds, JA could not be detected and the free-living mycelium of Tulasnella sp. (S6) featured a low JA content (1.63 ng/g.FW), but JA peaked most in the early germination stage (stage 2) in AG (Table 1). Further, all five kinds of hormones were detected in free-living mycelium of mycorrhizal fungus Tulasnella sp., albeit their context ranged almost 10-fold (0.44~4.29 ng/g) (Table 1). Please see page 3, lines 115-122.

  1. Figures 1, 3: the panels are very small and the observations described are hard to see

We tried to improve the resolution by change the font and the size. It looks like better than before. Please see Figure3 R1. Thanks.

  1. L204 what is SF?

Sorry, we are careless, It should be SG (symbiotic germination). we corrected it in the revised version. Thanks.

  1. L373 onwards and elsewhere: Make it a bit clearer what you mean here by nodulation. It is an analogy. Orchid roots do not nodulate.

Thanks very much for your helpful comments. Yes, You are right. We agree that orchid roots do not form nodule. Here, We mean that” based on our previous RNA-seq and iTRAQ data, we found four proteins encoding genes involved in the common symbiotic signal pathway, including two encoding nodulation signaling pathway protein (DoNSP2-1 and DoNSP2-2)” .Here, “the nodulation signaling pathway protein” refers our genes were functional annotated in NR database as nodulation signaling pathway protein which these proteins probably involved in symbiosis between nitrogen fixing bacteria (Rhizobium spp.) and Legum. In this sentence, we would like to express that the genes encoding two proteins were found in our transcriptomic data and we suspected they probably involved in orchid mycorrhizal formation because they were functional annotated as nodulation signaling pathway protein (it is known as key proteins for the formation of nodule in Legum and nitrogen fixing bacteria ).

So, we revised accordingly the sentence as “Based on our previous RNA-seq and iTRAQ data, we found four proteins encoding genes involved in the common symbiotic signal pathway, including two genes functional annotated as nodulation signaling pathway protein (DoNSP2-1 and DoNSP2-2) and two Ca2+ signal-related proteins, a calcium-dependent protein kinase (DoCDPK26), and a calmodulin-like protein (DoCML19). ”Please page 7, lines 382-386.

“We also found that the expression of genes encoding probably mycorrhizal signaling pathway proteins (DoNSP2-1 and DoNSP2-2) (functional annotated as nodulation signaling pathway proteins), both of which encode GARS-family transcriptional regulators, considerably increased under 0.5μM exogenous GA3 treatment in SG compared to AG. This result suggests exogenous GA3 probably affected the mycorrhizal specific gene expression by controlling the mycorrhizal-signaling pathway. Gibberellin’s ability to govern the nodulation signaling pathway in Lotus japonicus has been clarified by Maekawa et al. [38], who found that exogenous application of biologically active GA3 inhibited the formation of infection threads and nodules; hence they suspected GA halted the nodulation signaling pathway downstream of cytokinin, possibly at NSP2, which is required for Nod factor-dependent NIN expression. Whether a similar situation, which GA inhibited the downstream gene expression of mycorrhizal signaling pathway, occurs in orchid mycorrhiza needs to be confirmed (or not) in a co-culture system of orchid seedlings with its mycorrhizal fungi.” please page 8,lines 404-415.

  1. The language is good to read and clear. In some cases, some polishing could improve it. Some examples are given below, although they are not completely covering the polishing needed and a general treatment of the text would be nice.

L36 … minerals nutrients …

L50 … of the embryo …

L77 replace comma by a second hyphen … symbiosis –

transducing ….

L81 an orchid

L85 hormones level

L116 … plant hormones during …

L143 “can up” not clear language

* We changed the ” can up” to “achieved to”

L174 and page 8: Ca2

L197 ??? … biosynthesis: After …

L270 … on their mechanisms of …

L278 … revealed that the mature …

* L280-282 Sentence is not clear, grammar is insufficient.

We rewrite the sentence as “A little GA3 was detected in the early germination stage of SG and AG group but the content is no significant difference between SG and AG group.” please see page6, lines 311-312.

L324 … by fungus is still unclear …

L333 … that GAs are …

L335 … where they suppressed …/

L361 the genes encoding …

L373 … found that the expression …

L395 delete the second “genes”

L397 … by mycorrhizal fungi…

L408 arbuscular

L432 gibberellic acid

L485 … analysis was performed …

L495 … concentration of GA3 …

Many thanks for your kind help. We accepted all correction performed by reviewer. Meanwhile, we have revised thoroughly the MS and tried our best to avoid the spelling/grammatical error. Before submission, we have sent the MS to a professional editor [Charlesworth ] (https://www.cwauthors.com/APS) for improving the English.

Reviewer 2 Report

Dear authors,

I think this study has made a great achievement with a clear logic in writing and data presentation. Only a few minor mistakes need to be addressed. I recommend accepting this paper after revision. Please find my suggestions below:

  1. In line 68-69, we can find both gibberellin and GA in the same sentence. Since you have used abbreviations in the article, you might need to make it consistent.
  2. Would you be able to give a description regarding how long does it take for seeds to develop to stage 2, 3, and 4 for SG and AG in section 2.1.
  3. Please indicate the length of the scale bar in Figure 1.
  4. The 3 of GA3 should be subscript in line 138, 139, 141 and 143. Please check through the article.
  5. In Table 1, the word “ungermination” should try to make it in the same line. You also wrote U0 as the abbreviation in the table legend. Please make it consistent.
  6. Please improve the format of Table 2. It is confusing.
  7. Please analyze the significant difference and label it on Figure 4 to 8 between treatments.
  8. In the line 425-426 of 4.1 plant material and growing conditions, I wonder why you used different medium for SG and AG experiments. Do you think it could be a factor to affect the gene expression?
  9. Since you used ultra-HPLC for hormone analysis, why don’t you just write UHPLC in line 435.
  10. As you described in line 438, 1 g of seeds were applied for extraction. Since the seeds of orchids are super tiny, I just wonder if your description is correct, because 1 g of orchid seeds is a lot.

Author Response

Respond to reviewers’ comments

August. 18, 2020

Manuscript Number: ijms-908896
Title: Symbiotic and Asymbiotic Germination of Dendrobium officinale (Orchidaceae) Respond Differently to Exogenous Gibberellins

According to editor and reviewers’ helpful suggestions, we have revised our manuscript point by point and uploaded the revision file in the submission system. Our specific responses to the comments are listed below. Original comments raised by the editor and reviewers are in italics, and our responses are in regular font.

Reviewer 2

I think this study has made a great achievement with a clear logic in writing and data presentation. Only a few minor mistakes need to be addressed. I recommend accepting this

paper after revision. Please find my suggestions below:

Thanks for your great encouragement.

  1. In line 68-69, we can find both gibberellin and GA in the same sentence. Since you have used abbreviations in the article, you might need to make it consistent.

Thanks for your correction. We carefully check this point, and corrected gibberellin to GA. We made it consistent all text.

  1. Would you be able to give a description regarding how long does it take for seeds to develop to stage 2, 3, and 4 for SG and AG in section 2.1.

Thanks for your suggestion. We added the information in the result section. “In our previous study, we experimentally demonstrated that the seed germination of D. officinale on the oatmeal agar (OMA) medium with fungi is faster than seeds germination on 1/2 MS medium without fungi[7]. D. officinale seeds usually takes 10 days to develop up to stage 2 in SG, compared 16 days in AG. After 2 weeks of sowing seeds, more than 50 % of the seeds formed the protocorm structures (stage 3) in SG while the protocorm formation at least took 3 weeks in AG. About 20 days, seeds in SG can develop seedling stage (stages 4) compare to 30 days in AG. It took approximately 5 weeks to finish the germination process in SG while at least two months in AG.” please page 3,lines 97-104.

  1. Please indicate the length of the scale bar in Figure 1.

Thanks, we added the scale bar in Figure 1(we used a other pictures with the symbiotic and asymbiotic germination). The scale bar=0.5 mm.

  1. The 3 of GA3 should be subscript in line 138, 139, 141 and 143. Please check through the article.

Thanks for your correction. Sorry for our careless. In the revision version, we carefully check the point like this in whole text.

  1. In Table 1, the word “ungermination” should try to make it in the same line. You also wrote U0 as the abbreviation in the table legend. Please make it consistent.

Thanks for your good suggestion. We revised it according to your suggestion in Table 1. We wrote the “U0” for ungermination in text too.

  1. Please improve the format of Table 2. It is confusing.

Thanks. We improve the format of Table 2 by changing it to Horizontal version.

  1. Please analyze the significant difference and label it on

Thanks very much. In our study, gene expression analysis were performed following the reference<Livak KJ and SchmittenTD. Analysis of relative gene expression data using real-time quantitative PCR and the 2(-Delta Delta C(T)) Method.Methods. 2001,25(4):402-408. We added the significant difference analysis and label it on the figure4-fingure8. Meanwhile, we also noted it in the figure legends. Please see Figure 4-Figure 8.

  1. In the line 425-426 of 4.1 plant material and growing conditions, I wonder why you used different medium for SG and AG experiments. Do you think it could be a factor to affect the gene expression?

Thanks for your good question. Yes,we also consider the factor when we designed the experiment. In fact, asymbiotic germination on 1/2MS medium is not ideal control but no better choice at present. In almost all published references about orchid symbiotic germination, the system (asymbiotic germination on 1/2MS medium and symbiotic germination on OMA medium) was used because seed couldn’t germination on medium only with water + Agar although inoclulated fungus(or only up to stage 1). So, scholars have to use the OMA medium (with little nutrition- 0.25 %) for symbiotic germination. The co-culture system looks like mature system in orchid seed germination studies. Although we couldn’t subtract the affections by different medium, we consider the the medium affection on gene expression is less than mycorrhizal fungus.However, in my opinion, It's worth studying about the role or mechanism of the different medium on orchid seed germination in future.

  1. Since you used ultra-HPLC for hormone analysis, why don’t you just write UHPLC in line 435.

Thanks, we corrected it in the revision version.

  1. As you described in line 438, 1 g of seeds were applied for extraction. Since the seeds of orchids are super tiny, I just wonder if your description is correct, because 1 g of orchid seeds is a lot.

Thanks for your kind remind. Here, we described wrongly it in the first version. In the revision version, we checked the original data and corrected “50-200 mg of fresh seeds or fungi were frozen in liquid nitrogen and homogenized with a lysis buffer.... please page 9,lines 470-472.

Finally, thanks very much for all reviewers’ helpful suggestions and kind efforts.

Sincerely,

Juan